# The Safety of Anti-SARS-CoV-2 Vaccines: Vigilance Is Still Required

**DOI:** 10.3390/jcm11051248

**Published:** 2022-02-25

**Authors:** Michel Goldman

**Affiliations:** Institute for Interdisciplinary Innovation in Healthcare, Université Libre de Bruxelles, 1180 Brussels, Belgium; mgoldman@i3health.eu

**Keywords:** COVID-19, SARS-CoV-2, vaccine, mRNA lymphoma, T cell, autoimmunity

## Abstract

The opinion I put forward in this paper is that attention must continue to be paid to clinical observations compatible with a detrimental effect of anti-SARS-CoV-2 in certain diseases of immunological nature. Using the example of the atypical thrombocytopenic thromboses caused by adenoviral-vector-based vaccines, I argue that usual post-marketing pharmacovigilance programs may fail in identifying very rare vaccine-related disorders. Since the robust protective immunity induced by mRNA vaccines is related to their distinct capacity to induce strong stimulation of T follicular helper cells, I suggest that the safety of mRNA vaccines should be further assessed by appropriately designed epidemiological and mechanistic studies focusing on lymphoproliferative and autoimmune diseases in which T follicular helper cells were found to play a key role.

Vaccines against the SARS-CoV-2 virus have proven to be very effective in preventing severe forms of the COVID-19 disease. While around 10 billion doses have been administered worldwide, available data indicate that their safety profile is excellent. Indeed, the most frequent benign reactions fade quickly, being similar to those that can be observed with other vaccines. Thus, the risk–benefit balance of anti-COVID-19 vaccination is undoubtedly positive and justifies maximizing vaccination coverage as widely as possible. However, immunization campaigns face reluctance or frank opposition from part of the population. The fear that certain adverse effects have not yet been identified is the main reason underlying this vaccine hesitancy. The concern is fueled by isolated observations for which the causal relationship is impossible to validate or discard via conventional post-marketing pharmacovigilance programs. The objective of this article is to encourage studies specifically designed to confirm or rule out the cause-and-effect relationship between vaccination and very rare complications. It will be essential to communicate in a transparent way on the information that will be collected, so as not to leave the monopoly of these subjects to the opponents of vaccination.

Indeed, a few rare, severe, adverse effects were already recognized by regulatory authorities and led to the adaptation of vaccination strategies. Anaphylactic reactions were the first to be easily identified as they develop within minutes after vaccine administration. Their incidence (around 5 per million shots) was found to be similar to that observed with other vaccines. Importantly, polyethylene glycol, which is present in adenoviral-vector-based and mRNA vaccines, was identified as the ingredient responsible for most of these allergic reactions [1]. The NVX-CoV2373 vaccine from Novavax is devoid of polyethylene glycol, and no anaphylactic reactions were reported with this product [2]. Therefore, patients hypersensitive to polyethylene glycol can be safely immunized with the NVX-CoV2373 vaccine.

The next serious complication to be brought to light was atypical thrombocytopenic thrombosis occurring after the administration of adenoviral-vector-based vaccines in young women. The consequences of these thromboses can be dramatic, especially when they occur in cerebral vessels. The cause-and-effect relationship was initially refuted based on the comparable incidence of classical thromboses in vaccinated and unvaccinated populations. Because a handful of clinicians across Europe were struck by the analogy with a rare complication of heparin treatment, the association of atypical thrombosis with anti-platelet factor 4 antibodies was established [3]. It was then formally recognized by regulatory agencies as a complication of both ChAdOx1nCoV-19 (AstraZeneca/Oxford University) and Ad26.COV2.S (Johnson & Johnson) vaccines. Targeted epidemiological studies allowed the risk to be evaluated at 2 cases per 100,000 doses below the age of 50 years. On this basis, regulatory agencies adapted their recommendations for the use of adenoviral vector vaccines according to age, avoiding exposing younger individuals to this very rare complication. More recently, Guillain–Barré syndrome was recognized as another very rare complication of adenoviral-vector-based vaccination. As the incidence of Guillain–Barré syndrome was 5 times higher following COVID-19 infection, this finding did not affect the overall positive risk–benefit balance of adenoviral-vector-based vaccines [4]. Myocarditis observed in the days following the injection of messenger RNA vaccine in young individuals is another well-established rare complication, first identified within the Israeli health system [5]. The higher risk with the mRNA-1273 (Moderna) vaccine as compared to the BNT162b2 (Pfizer/BioNTech) vaccine [6] led several countries to only use the BNT162b2 vaccine in younger individuals and to cut the dose of the mRNA-1273 vaccine by half when given as a booster. 

Although the likelihood that other rare side effects might be revealed is low, it is important to track them for several reasons. First and foremost, if at-risk subjects can be identified, they might benefit from an adapted vaccine strategy. Secondly, it is important to accumulate information to refute false allegations but also to provide fair compensation to victims of plausible vaccine-related damages. Thirdly, continuous monitoring and transparent reporting of side effects is essential to gain the trust of the wider public. In Europe, the marketing authorization provided by the European Medicines Agency for COVID-19 vaccines is indeed subject to annual renewal so that appropriate action can be taken if needed [7].

Herein, we suggest that still unidentified, rare complications of mRNA vaccination might result from the distinct capacity of lipid nanoparticle-encapsulated, nucleoside-modified mRNA vaccines to hyperactivate the T follicular helper cells (TFHs), which are essential for the formation of germinal centers [8]. Indeed, vaccines based on this technology were shown to be much more efficient than adjuvanted protein or inactivated viral vaccines to induce TFH responses in mice [8]. A recent study in human volunteers confirmed that TFHs specific for SARS-CoV-2 peptides persist in draining lymph nodes up to 200 days following primary vaccination [9]. The hyperactivation of FTH leading to potent germinal center responses explains the transient reactive lymphadenopathy which occasionally develops in the region draining the injection site of a SARS-CoV-2 mRNA vaccine. When a lymph node biopsy was required for differential diagnosis, the pathological hallmark was follicular hyperplasia with prominent germinal centers [10].

Since TFHs are key players in some T-cell malignancies and certain autoimmune diseases [11], one may wonder whether TFH hyperactivation induced by SARS-CoV-2 mRNA vaccines might influence the course of these disorders. Our recently published observation of the rapid progression of angioimmunoblastic T-cell lymphoma (AITL) following a booster shot of the BNT162b2 vaccine suggests that this question needs to be considered [12]. Indeed, the current understanding of the pathogenesis of AITL allows one to formulate a sound hypothesis on the possible effect of the messenger RNA vaccine in this clinical setting. Recent studies have identified mutations in genes of bone marrow stem cells of some AITL patients, suggesting that a premalignant condition named clonal hematopoiesis might represent the starting point of their disease [13]. When these mutated stem cells differentiate into T lymphocytes, they are subject to other mutations promoted by epigenetic mechanisms. One of the key mutations is in the RHOA gene. Combined with the TET2 mutation already present in stem cells, the RHOAG17V mutation is sufficient to induce in mice lymphomatous lesions which mimic AITL in all respects [14]. In both humans and animals, the essential characteristic of neoplastic TFHs is that they proliferate very actively following minimal stimulation by dendritic cells [14]. The neoplastic TFHs of the patient we described carry both RHOAG17V and TET2 mutations, which led us to suggest that their stimulation by the BNT162b2 vaccine could explain the outbreak of the lymphoma. Ongoing experiments in mice should confirm or refute this hypothesis. Intriguingly, another case of T-cell lymphoma recurrence following the administration of the BNT162b2 vaccine was recently reported [15]. Although the mRNA vaccine might have played a pathogenic role in our patient, it is still possible that the lymphoma flare would have occurred in absence of vaccination. Additional studies focusing on patients with pre-existing somatic mutations might be necessary to further explore the cause-and-effect relationship that we suggest.

Among the possible impact of mRNA vaccines on other diseases of immunological origin, attention should also be paid to type 1 diabetes, since TFHs were shown to play a key role in the development of autoantibodies attacking cells in pancreatic islets [16]. Furthermore, the blood of children at high genetic risk for diabetes contains high numbers of activated TFH even before they develop hyperglycemia [17]. Importantly, natural SARS-CoV-2 infection also results in the differentiation of TFHs [18]. Therefore, COVID-19 itself might also represent a risk factor for certain types of TFH-mediated disorders, including type 1 diabetes. Indeed, SARS-CoV-2 infection was shown to be associated with an increased incidence of type 1 diabetes [19].

It is very unlikely that existing pharmacovigilance programs will be able to establish the risk of SARS-CoV-2 vaccination revealing or accelerating the course of lymphomas or other immunological diseases. As previously mentioned, we should remember that the thrombotic thrombocytopenia rarely induced by the ChAdOx1nCoV-19 vaccine was not primarily identified by pharmacovigilance systems but came to light because of a few isolated case reports, followed by a series of mechanistic studies [2]. Likewise, studies specifically designed and powered to investigate possible rare TFH-related reactions suggested by well-documented case reports and a sound scientific rationale might allow detrimental consequences of SARS-CoV-2 vaccination to be prevented in patients with certain lymphoproliferative or autoimmune diseases. Such an approach might also help us understand recent surprising observations of cancer remission following SARS-CoV-2 mRNA vaccination [20]. We believe that careful post-vaccination surveillance based on the best science will help to build trust and reduce vaccine hesitancy.

## Data Availability

Not applicable.

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
