# Peer review of "The Safety of Anti-SARS-CoV-2 Vaccines: Vigilance Is Still Required"

_jcm, 2022, doi:10.3390/jcm11051248_

Round 1

Reviewer 1 Report

The points raised by the author are a valuable contribution to the scientific discussion. However, two points could be considered by the author:

1) The author proposes a hypothesis that rare malignant diseases like AITL might be caused by Covid-19 vaccines. It would be reasonable to include calculations on the frequency of such conditions and thus estimate the chance of a coincidental occurrence of AITL after vaccination. I don't mean an exact calculation with numbers here, but the possibility should be mentioned that these individual cases were coincidental correlation rather than causation.

2) It is discussed in detail how Covid-19 vaccines induce TFH in mice and in humans. It would also be of interest to evaluate the influence that SARS-CoV-2 infection has of TFH, in order to compare the risk of vaccination vs. infection.

Minor Comments / suggestions according to line:

11 – remove the word „humoral”: “protective immunity” is a clinical category (i.e. protection from disease) and as the author alludes to further in the text, not only due to humoral responses.

45 – “occur in” instead of “enter”

80 – “peptides” plural

Author Response

Thank you very much for your thoughtful comments and suggestions.Your two points are now addressed in the paper.
1. I made clear that “Although the mRNA vaccine might have played a pathogenic role in our patient, it is still possible that the lymphoma flare would have occurred in absence of vaccination. Additional studies focusing on patients with preexisting somatic mutations might be necessary to further explore the cause-and-effect relationship that we suggest.” ;

2. I also add a sentence and a reference about TFH during COVID-19 disease. "Importantly, the natural SARS-CoV-2 infection also results in the differentiation of FTH (17). Therefore, COVID-19 itself might also represent a risk factor for certain types of FTH-mediated disorders including type 1 diabetes."

Minor corrections were made as suggested.

Reviewer 2 Report

This manuscript is an Opinion article which aims to highlight the importance of continuous monitoring of detrimental clinical effects of anti-SARS-CoV-2 vaccines among patients with possible immunological complications. It primarily discussed the reported safety profiles and side effects of different COVIV-19 vaccines available thus far, while discussing the lesser known effects of mRNA vaccination towards the development of malignancies and diabetic complications. The author should be credited for this interesting yet impactful Opinion article that contains important argument points which could encourage a wider proportion of the general population towards obtaining their vaccination. I have a few points that would be helpful in improving the manuscript further:

1) Anaphylactic complications are among the most commonly reported side effects of the COVID-19 vaccines, and the author highlighted that the polyethylene glycol-free Novavax vaccine (NVX-CoV2373) is an important step forward for patients hypersensitive to polyethylene glycol. I felt that the author can discuss further on the available safety profiles and efficacies of this vaccine. The author can add a recently published article as follow regarding the safety profiles and efficacies of the Novavax vaccine (NVX-CoV2373).

Dunkle, Lisa M., Karen L. Kotloff, Cynthia L. Gay, Germán Áñez, Jeffrey M. Adelglass, Alejandro Q. Barrat Hernández, Wayne L. Harper et al. "Efficacy and Safety of NVX-CoV2373 in Adults in the United States and Mexico." New England Journal of Medicine (2021).

2) While continuous post-vaccination surveillance is crucial in evaluating the safety profiles of SARS-CoV-2 vaccines, the vaccination coverage and acceptance rates are among the crucial factors in determining the success rate of a vaccination program within a community. The author may discuss further on this point as these are all connected in a cycle: More useful and accurate information from post-vaccination surveillance would help in reducing vaccination reluctancy, hence higher vaccination rates, more data and subjects for post-vaccination surveillance.

3) Line121: "TFH" was spelled as "FTH".

Author Response

Thank you very much for your thoughtful comments that are addressed in the revised version:

  1. Indeed the NEJM paper which is now cited clearly state that the Novavax vaccine does not elicit anaphylactic reactions.
  2. We ended the paper by the following sentence: 

    We believe that careful post-vaccination surveillance based on the best science will help to build trust and reduce vaccine hesitancy.